# Effect of Antihypertensive Drug (Chlorothiazide) on Fibrillation of Lysozyme: A Combined Spectroscopy, Microscopy, and Computational Study

**DOI:** 10.3390/ijms24043112

**Published:** 2023-02-04

**Authors:** Nojood Altwaijry, Ghaliah S. Almutairi, Mohd Shahnawaz Khan, Gouse M. Shaik, Majed S. Alokail

**Affiliations:** Department of Biochemistry, College of Science, King Saud University, Riyadh 11451, Saudi Arabia

**Keywords:** Chlorothiazide, Lysozyme amyloidosis, aggregated HEWL, fluorescence, Molecular Modeling

## Abstract

Amyloid fibrils abnormally accumulate together in the human body under certain conditions, which can result in lethal conditions. Thus, blocking this aggregation may prevent or treat this disease. Chlorothiazide (CTZ) is a diuretic and is used to treat hypertension. Several previous studies suggest that diuretics prevent amyloid-related diseases and reduce amyloid aggregation. Thus, in this study we examine the effects of CTZ on hen egg white lysozyme (HEWL) aggregation using spectroscopic, docking, and microscopic approaches. Our results showed that under protein misfolding conditions of 55 °C, pH 2.0, and 600 rpm agitation, HEWL aggregated as evidenced by the increased turbidity and Rayleigh light scattering (RLS). Furthermore, thioflavin-T, as well as trans electron microscope (TEM) analysis confirmed the formation of amyloid structures. An anti-aggregation effect of CTZ is observed on HEWL aggregations. Circular dichroism (CD), TEM, and Thioflavin-T fluorescence show that both CTZ concentrations reduce the formation of amyloid fibrils as compared to fibrillated. The turbidity, RLS, and ANS fluorescence increase with CTZ increasing. This increase is attributed to the formation of a soluble aggregation. As evidenced by CD analysis, there was no significant difference in α-helix content and β-sheet content between at 10 µM CTZ and 100 µM. A TEM analysis of HEWL coincubated with CTZ at different concentrations validated all the above-mentioned results. The TEM results show that CTZ induces morphological changes in the typical structure of amyloid fibrils. The steady-state quenching study demonstrated that CTZ and HEWL bind spontaneously via hydrophobic interactions. HEWL–CTZ also interacts dynamically with changes in the environment surrounding tryptophan. Computational results revealed the binding of CTZ to ILE^98^, GLN^57^, ASP^52^, TRP^108^, TRP^63^, TRP^63^, ILE^58^, and ALA^107^ residues in HEWL via hydrophobic interactions and hydrogen bonds with a binding energy of −6.58 kcal mol^−1^. We suggest that at 10 µM and 100 μM, CTZ binds to the aggregation-prone region (APR) of HEWL and stabilizes it, thus preventing aggregation. Based on these findings, we can conclude that CTZ has antiamyloidogenic activity and can prevent fibril aggregation.

## 1. Introduction

The folding of proteins and peptides plays a key role in maintaining life’s normal functioning. Incorrect folding may cause the protein to lose its original activity and function, or even produce potentially damaging protein aggregates. Initially, the protein forms monomers with different folding degrees, resulting in highly disordered, partially structured, or native-like oligomers. Amyloid forms more stable species that are dense, larger, and have a β-sheet structure. These proteins then grow to form fibers with a cross-β structure, which can accumulate in the body organs [1]. This abnormal perception of protein is responsible for most amyloid-related diseases, such as Alzheimer’s disease, type II diabetes disease, Parkinson’s syndrome, familial amyloidosis, and Huntington’s disease [2].

Human lysozyme (HL) is a bacteriolytic enzyme found in body fluids like saliva and tears, as well as tissues like cartilage, liver, and kidney. The egg white is also considered to be a rich source of lysozyme, in addition to human fluids and tissues [3]. Lysozyme amyloidosis (ALys) is a rare type of systemic amyloidosis. It is caused by a single point mutation in the lysozyme gene. To date, 10 amyloid point mutations have been reported [4]. Since lysozymes are widely distributed throughout the body, ALys can cause numerous organ disorders, such as digestive damage, spontaneous liver rupture, skin mucosal disease, heart failure, renal dysfunction, repeated pulmonary infectious episodes, and granulomatosis of the bronchi [5,6,7]. The most extensively studied lysozyme to form fibrils and aggregates is HEWL [8]. This globular protein consists of 129 amino acid residues and 4 disulfide bridges that make up a single folded polypeptide chain (Figure 1A). HEWL is structurally homologous to HL. Both of them have α and β domains and four disulfide bonds. The α-domain comprises 4 helices: A-helix, B-helix, C-helix, and D-helix. Whereas, the β-domain is composed of central 3_10_ helices, a large loop, and a triple-stranded β-sheet. Both HL and HEWL have glycosidase activity in which they cause hydrolysis of the 1,4 glycosidic bonds between N-acetylglucosamine and N-acetylmuramic acid in peptidoglycan in bacterial cell walls [9]. Their only difference is that HL has an extra VAL^130^ residue at the N-terminus. Furthermore, it was demonstrated that under certain conditions HEWL forms in vitro fibrils that are identical to those of HL [8,10,11]. The availability and solubility of HEWL in aqueous media combined with its small size make it an ideal model for the study of lysosomal aggregation and amyloidosis [11]. Protein amyloidosis has gradually become a hot research topic despite considerable efforts to treat the problem. Substantial efforts have been directed at identifying inhibitors of amyloid aggregation, which remains a significant threat to human health.

Thiazide diuretics have been used as oral antihypertension drugs for more than 60 years. There are two major classes of thiazide diuretics: thiazide-like diuretics (lack the thiazo ring) and thiazide-type diuretics (have the thiazo ring) [2]. During kidney function, thiazide diuretics elevate water excretion and inhibit Na^+^ reabsorption in the distal convoluted tubule lowering blood pressure. The thiazide diuretics also act as vasodilators by lowering renin levels, which prevents the conversion of angiotensinogen to angiotensin I, preventing arteriolar vasoconstriction and lowering blood pressure [12,13]. Launer et al. found that high blood pressure is associated with Alzheimer’s disease [14]. Chou et al. suggest that hypertension drugs can reduce the incidence and progression of Alzheimer’s disease [15]. The study by Hajjar et al. found a positive effect of antihypertensive medications on Alzheimer’s disease patients with hypertension [16]. Previous studies have shown that inhibiting angiotensin-converting enzyme (ACE) activity reduces the formation of Aβ plaques and neurodegeneration [17]. Zhao et al. demonstrate that valsartan prevents oligomerization of the Aβ peptide in vitro [18]. Further studies have shown that valsartan enhances cognitive function and reduces amyloid beta (Aβ) aggregation in mice [15]. Angiotensin II receptor blockers (ARBs) have been shown to delay the onset of Alzheimer’s disease [19]. The results of the animal study show that diuretics can reduce the aggregation of amyloid beta 1–42 (Aβ_1–42_) in the brain [20]. It was also found that thiazides among different antihypertensive medications reduced the risk of Alzheimer’s disease [21]. According to epidemiological studies, Alzheimer’s disease develops later in life, is related to high midlife blood pressure, and hypertension may contribute to cognitive decline and dementia. Thiazides, therefore, have been shown to reduce AD risk among different classes of antihypertensive medications [21]. Diuretics, such as triamterene, spironolactone, and amiloride, have also been reported to have neuroprotective effects associated with better verbal learning and memory [22]. Furthermore, indapamide was reported to suppress the production of Aβ and promote its degradation, which results in decreased Aβ amyloid aggregation [2]. In addition, another previous study has reported the ability of indapamide and hydrochlorothiazide to suppress human serum albumin (HSA) and human lysozyme (HL) aggregation [2].

**Figure 1 ijms-24-03112-f001:**
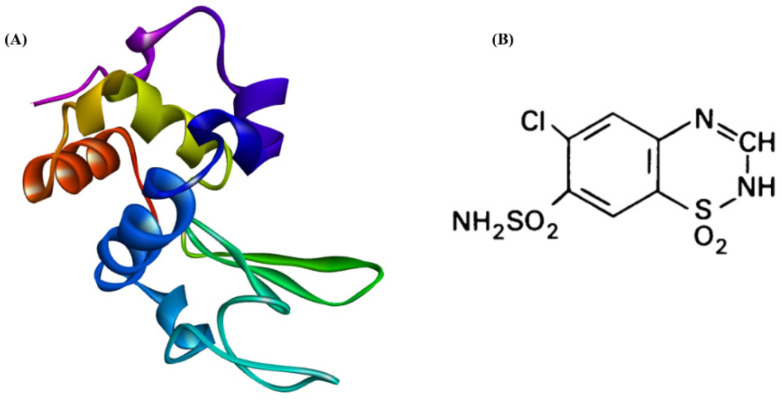
(**A**) Folded structure of HEWL. (**B**) Chemical structure of CTZ [23].

This study was conducted to investigate the antiaggregation effects of chlorothiazide (CTZ) diuretic on HEWL aggregation (Figure 1B).

## 2. Materials and Methods

### 2.1. Materials

MedChemExpress (MCE) provided the CTZ (Monmouth Junction, NJ 08852, USA). Sigma-Aldrich provided HEWL, 8-anilino-1-naphthalenesulfonic acid (ANS), Thioflavin-T (ThT), and the rest of the chemicals (St. Louis, MO 68178, USA). All experimental solutions were made to an analytical grade in double-distilled water filtered with the Milli-Q system. The pH was measured with a SENTRON INTEGRATED SENSOR TECHNOLOGY 2001 pH meter (Texas City, TX, USA). Agilent Technologies Cary Eclipse-Fluorescence Spectrophotometer was used for all fluorescence measurements (Santa Clara, USA).

### 2.2. Preparation of HEWL Fibrils

HEWL samples of 100 µM were prepared in glycine buffer, pH 2.0 (100 mM glycine/HCl, 100 mM sodium chloride) and filtered through a 0.45 µm filter syringe with and without CTZ (10 µM & 100 µM). HEWL’s concentration at 280 nm was calculated using a molar extinction coefficient of 37,970 M^−1^cm^−1^ [23]. To prepare CTZ, dimethyl sulfoxide (DMSO) was used and then filtered through a 0.45 µm filter syringe. A Milli-Q solution was used to dilute the CTZ stock solution to the desired concentration. We incubated HEWL samples at 55 °C, 600 rpm for 10 h to induce in vitro production of amyloid fibrils. HEWL solutions were aliquoted at different time points after being vortexed to ensure that amyloid fibrils were well distributed [24]. All experiments were performed at 10 µM of HEWL.

### 2.3. Turbidity Analysis

The formation of amyloid aggregates can be studied by measuring turbidity at 350–450 nm. Turbidity measurement can only detect amyloid fibrillation since protofibrils are too small to reflect 400 nm light. These measurements, however, require relatively high protein concentrations [24]. In this study, HEWL turbidity measurements in the presence and absence of 10 µM and 100 µM CTZ were performed using a Perkin Elmer Lambda 25 double beam spectrometer in a cuvette at 350 nm absorbance with 1 cm path length. Furthermore, the formation of HEWL fibrils was monitored over 10 h in the absence of CTZ.

### 2.4. Rayleigh Light Scattering (RLS) Measurement

Rayleigh Light Scattering (RLS) measurements of fibrillated HEWL were conducted at room temperature with and without 10 µM and 100 µM CTZ. After the protein was excited at 350 nm, its emission spectra were recorded in the range of 300 nm to 600 nm. Both excitation and emission were performed with a 2.5 nm slit width. The RLS was used to detect aggregation pathways over 10 h in absence of CTZ [24].

### 2.5. ThT Fluorescence Assay

In Milli-Q, a stock solution of ThT was prepared and then filtered through 0.45 μm filters. ThT concentration was calculated using a molar extinction coefficient of 36,000 M^−1^cm^−1^ for 412 nm absorption [24]. The ThT fluorescence measurements were conducted on 10 µM HEWL samples incubated at pH 2 and 55 °C for 10 h in the presence and absence of CTZ (10 & 100 µM). The kinetics of HEWL fibrillation was determined by collecting aliquots of HEWL incubation mixtures every 2 h for up to 10 h in the absence of CTZ. Following that, HEWL samples were diluted with glycine/HCl buffer up to 10 µM. A 1:1 molar ratio of ThT was then added to HEWL samples in the presence and absence of CTZ and incubated in the dark for 30 min at 25 °C. The excitation wavelength was 440 nm and emission wavelengths ranged from 450 to 600 nm with a 5 nm slit width [23].

### 2.6. ANS Binding Assay (Hydrophobicity Analysis)

Hydrophobic patches can be detected on proteins’ surfaces using the ANS binding assay. To prepare the stock solution, ANS was dissolved in ethanol and its concentration was determined by measuring the absorbance at 350 nm using a molar extinction coefficient of 5000 M^−1^ cm^−1^ [23]. Before measuring ANS fluorescence, 10 µM of each HEWL sample native, fibrillated, and in the presence of CTZ were added to a 50-fold molar excess of ANS and incubated further in the dark for 30 min at room temperature. Next, HEWL samples were excited at 380 nm and emissions were recorded between 400 and 600 nm with a slit width of 5 nm [23].

### 2.7. Far-UV Circular Dichroism

To assess the secondary structures of HEWL during the aggregation process, Far-UV CD measurements were conducted with an Applied Photophysics Chirascan Plus spectrometer. We have scanned CD spectra in the range of 190–250 nm in a cuvette with a path length of 0.1 mm. The experiment was conducted with 10 µM HEWL in the presence and absence of 10 µM and 100 µM of CTZ. Calculation of the secondary structure was done using the BeStSel tool.

### 2.8. Steady State Measurement

To study the binding relationship between HEWL and CTZ, a steady-state fluorescence quenching measurement was performed on HEWL at three different temperatures (298 K, 303 K, and 310 K). A 5 µM HEWL concentration was fixed, and the 1 mM CTZ volume was increased based on the degree of quenching. Fluorescence intensity was measured by exciting the protein at 295 nm and recording its emission spectrum at 300–400 nm. The HEWL solution was prepared in 20 mM sodium phosphate buffer at pH 7.4, while a CTZ solution was prepared in absolute DMSO and then diluted to the appropriate concentration using Milli-Q. The Stern–Volmer Equation (1)) was used to analyze the data [23].
(1)$$\frac{F_{0}}{F}=1+K_{SV}{\left[Q\right]}_{0}=1+K_{q}\tau_{0}{\left[Q\right]}_{0}$$
where $F_{0}$ and F represent fluorescence intensities of HEWL with and without chlorothiazide, ${\left[Q\right]}_{0}$ is CTZ concentration, $K_{q}$ represent the bimolecular rate constant of the quenching reaction, $K_{SV}$ is the Stern–Volmer quenching constant, and $\tau_{0}$ is the fluorescence lifetime of protein, which is ~10^−8^ sec. Binding constants and binding sites will be obtained from the Equation (2): [23]
(2)$$\log\;[(F_{0}-F)/F]=\log\;K_{b}+n\;\log\;{\left[Q\right]}_{0}$$
where n reflect the number of binding sites and $K_{b}$ represent the binding constant. Change in entropy (ΔS°), enthalpy (ΔH°), and Gibbs free energy (ΔG°) was calculated using Van’t Hoff equation (Equation (3)) and Gibbs free energy equation (Equation (4)), respectively:
(3)$$\ln K_{b}=-\frac{\Delta H{}^{\circ}}{RT}+\frac{\Delta S{}^{\circ}}{R}$$
(4)$$\Delta G{}^{\circ}=\Delta H{}^{\circ}-T\Delta S{}^{\circ}=-RT\ln K_{b}$$
where *R* represent the gas constant, which is equal to 1.987 cal mol^−1^ K^−1^. T is the absolute temperature (K).

### 2.9. Transmission Electron Microscopy (TEM)

Incubation of HEWL at pH 2.0, 55 °C and with and without 10 µM and 100 µM CTZ for 10 h was observed using a JEOL JEM1400 Transmission Electron Microscope with 120 kV accelerating voltage. After dropping the samples, a carbon-stabilized formvar film was applied over a 300-mesh copper grid and any excess fluid was removed after 2 min. After that, the grids were negatively stained with uranyl acetate. Images were viewed at various magnifications.

### 2.10. Molecular Modeling Studies

#### 2.10.1. Molecular Docking

With AutoDock 4.2.6 and NAMD [1], molecular docking and dynamic simulations have been conducted to explore the interaction between CTZ and HEWL [25,26]. HEWL (6LYZ) was retrieved from the Protein Data Bank (PDB) and CTZ (CID: 2720) was obtained from PubChem [26]. All hydrogen atoms were added while all water molecules were removed. Afterward, Kollman charges were added to the protein and the protein was made to be rigid. For each site, 0.375 Å grid spacing was used as grid size. A 150-papulation size in GA was used. Following that, molecular dynamics simulations were performed for 5 ns on all predicted sites. Discovery Studio 3.5 was used to visualize and identify binding residues.

#### 2.10.2. Molecular Dynamic Simulation

Molecular dynamics (MD) simulations of the free HEWL and its docked complex with CTZ were performed using the Charmm-GUI and NAMD packages. Molecules were solvated in a cubic box with water molecules using the NAMD force field. The rectangular box was generated with a system size of 67 Å for A, B, and C. The periodic boundary was 90 Å α, β, and γ. All the ions were neutralized before the simulation. CTZ was restrained at 300 K and equilibrated at NVT (constant number, volume, and temperature) followed by NPT (constant number, pressure, and temperature), at 1 bar and 300 K, respectively. The production simulations were performed at 300 K for 5 ns after equilibration. The conformational changes in the free HEWL and its docked complex with CTZ were assessed by computational analysis via root-mean-square deviation (RMSD), radius of gyration (Rg), and root-mean-square fluctuation (RMSF).

#### 2.10.3. Statistical Analysis

The statistical analysis was conducted using SPSS. One-way ANOVA was used to determine the statistical significance of the experimental results. Statistics are considered significant when they have a *p* value of ^#^
*p* ≤ 0.05, ^##^
*p* ≤ 0.01 or ^###^
*p* ≤ 0.001.

## 3. Results and Discussion

### 3.1. Rayleigh Scattering Measurement (RLS)

In this study, RLS was used to test the antiaggregation effect of CTZ against the fibrillation of HEWL by measuring fluorescence at 350 nm [27]. HEWL was incubated at pH 2.0, 55 °C, and 600 rpm for 10 h to induce amyloid fibrillation. Native HEWL at 25 °C exhibit little scattering indicating the absence of amyloid fibrils. Different results were observed when HEWL was coincubated with 10 µM and 100 µM of CTZ regarding aggregation. The RLS peak reflects the presence of aggregation and prescription. The effects of various CTZ concentrations on HEWL fibrils are illustrated in Figure 2. The RLS is dramatically reduced by 67% at 10 µM CTZ. However, at 100 µM CTZ the RLS is dramatically decreased by 63%. Both concentrations seem to reduce light scattering, but CTZ at 10 µM likely reduces RLS, hence aggregation more effectively than 100 µM CTZ. This pattern could be attributed to the ability of diuretics to form protofibrils or amyloid aggregates of HEWL [28]. It appears, however, that CTZ has antiaggregation effects against the HEWL aggregate.

### 3.2. Turbidity Measurement

Turbidity measurement is a common technique to detect protein aggregation and large precipitations [24]. In this study, the formation of HEWL aggregation was assessed using turbidity measurements at 350 nm [23]. The increase in absorbance reflected aggregate formation, which was most apparent (precipitates) and noticeable after 8 h (Figure 3A). The effects of CTZ on amyloid aggregation were also examined by measuring HEWL aggregate turbidity in the presence and absence of 10 µM and 100 µM CTZ. Figure 3B indicates that CTZ significantly decreases the 8 h aged HEWL aggregates. At both tested concentrations of CTZ, turbidity was decreased when compared to aggregated HEWL. This suggests that CTZ significantly interferes with the formation of amyloid fibrils.

### 3.3. ThT Fluorescence Assay

Amyloid fibrillation is characterized by the binding of ThT dye to β-sheet structures [29]. ThT fluorescence assays are effective for detecting protein fibrillation and quantifying amyloid aggregation [5]. The presence of fibrils of amyloid is reflected by an increase in fluorescence intensity. Figure 4A shows the aggregation kinetics of the HEWL fibril formation pathway. ThT fluorescence of HEWL increases over time due to the formation of amyloid fibrils. As shown in Figure 4A, HEWL aggregation follows a sigmoidal curve consisting of a lag phase, an elongation phase, and a stationary phase. The intensity of ThT fluorescence in native HEWL does not demonstrate any significant increases in intensity when compared to 8 h aged HEWL due to the lack of β-sheets (Figure 4B). Fibrils of HEWL have a maximum peak at 485 nm [30]. To explore whether CTZ affects HEWL fibrils growth, we measured their fluorescence intensities at 10 µM and 100 µM of CTZ. Figure 4B illustrates the effect of CTZ against HEWL aggregation. In the presence of 10 µM and 100 µM CTZ, the ThT intensity and amyloid fibril content were diminished by 76% and 74%, respectively.

### 3.4. Effect of CTZ on Surface Hydrophobicity of HEWL

ANS fluorescence assay allows the identification of protein folding intermediates and the presence of hydrophobic patches on the surface [31]. Increased fluorescence intensity is triggered by the binding of ANS dye to hydrophobic groups. Native HEWL has hydrophobic residues that are buried inside the folded conformation, resulting in negligible fluorescence intensity. As contrasted with fibrillated HEWL, there was an increase in fluorescence intensity due to the presence of hydrophobic residues [32]. To investigate the effect of CTZ on the hydrophobicity of the protein surface, ANS fluorescence measurements were performed. Nonfibrotic HEWL samples generally exhibit lower ANS fluorescence intensity than HEWL samples with amyloid fibrils [5]. It has been observed that CTZ affects the surface hydrophobicity of HEWL in a concentration dependent manner. Compared to the fibrillated HEWL, both CTZ concentrations significantly reduced the exposure to hydrophobic patches [23] (Figure 5). It was found, however, that the number of hydrophobic patches that were exposed to the surface was less in the presence of 10 µM CTZ than in the presence of 100 µM CTZ. Together, these results indicate fewer exposed hydrophobic patches of HEWL in the presence of 10 µM CTZ. Several studies also suggest that ANS can bind to the external sites of proteins, creating ion pairs [33]. Thus, more biophysical approaches are needed to understand CTZ behavior in the presence of HEWL amyloid aggregates.

### 3.5. Circular Dichroism (CD) Measurement

Measurements of CD spectroscopy are extensively used in proteomic studies to study changes in the secondary structure of proteins [34]. Far-UV CD measurements were used to determine the changes in the secondary structures of HEWL after binding with CTZ. In the native HEWL, the typical α-helix structure was determined at 208 nm and 222 nm. HEWL fibrillated with a β-sheet structure shows a negative band at 218 nm CD spectrum as well as a positive ellipticity at 196 nm [35]. According to Figure 6A, it was found that the proportion of β-sheets decreased significantly, and the number of α-helices increased after co-incubating HEWL fibrils with both CTZ concentrations. Furthermore, no significant difference was observed in β-sheets content between 10 µM and 100 µM of CTZ. Figure 6B illustrates the percentage of secondary structure changes in the presence and absence of CTZ. From our far-UV CD data, we conclude that the addition of CTZ can effectively prevent the α-to-β transition of HEWL at both concentrations.

### 3.6. Fluorescence Quenching Analysis

The interaction of small molecules, such as drugs and biological macromolecules, can be assessed by fluorescence quenching spectroscopy [36]. The conformational changes that occur upon drug binding are studied using the aromatic amino acids tyrosine, phenylalanine, and tryptophan. Our study used this approach to study the interaction between HEWL and CTZ at three different temperatures (298 K, 303 K, and 310 K) to calculate thermodynamic parameters. According to Figure 7A, the protein was excited at 295 nm and the emission spectra ranged from 300 nm to 400 nm. An emission peak was observed at 345 nm and gradually decreased after adding CTZ. A gradual decrease in fluorescence intensity was observed as the concentration of CTZ was increased until 27.5 µM to a fixed 5 µM HEWL indicating an interaction between CTZ and HEWL.

#### 3.6.1. Binding Affinity and Mechanism

The Stern–Volmer equation was used to calculate the $K_{q}$ and $K_{SV}$ values of the HEWL–CTZ interaction (Figure 7B). The quenching phenomenon was determined using the $K_{q}$ value [5]. Table 1 demonstrates that the values of quenching rate constant $K_{q}$ and the Stern–Volmer quenching constant $K_{SV}$ increased as the temperature increased, suggesting the dynamic quenching mechanism may be responsible for the decreasing in HEWL–CTZ fluorescence emission. Dynamic quenching derived from molecule colliding, in which the number of colliding molecules increases, thereby increasing energy transfer and thus increasing the quenching constant of the fluorescent compound as temperature increases [36]. $K_{b}$ is a measure of the stability of the HEWL–CTZ complex. The higher the value, the stronger the affinity. It appears that the CTZ–HEWL interaction is more stable at high temperatures since the $K_{b}$ value rises with temperature [26].

#### 3.6.2. Thermodynamic Parameters

Gibbs free energy (ΔG°), entropy (ΔS°), and enthalpy (ΔH°) are thermodynamic parameters that play a crucial role in controlling the interaction of a protein with a drug. Their measurements can assist in better understanding the forces that contribute to the formation of the HEWL–CTZ complex [5]. Electrostatic interaction is indicated by low or negative values of ΔH° with positive ΔS°. Negative values of ΔH° and ΔS° are linked with hydrogen bonding and van der Waals interaction [37]. Moreover, positive ΔH° and ΔS° values indicate the presence of hydrophobic interactions [38]. As a result of the CTZ–HEWL interaction, the entropy of the system changes during the initial attachment of the HEWL to CTZ. Various forces, including hydrophobic interactions, van der Waals interactions, hydrogen bonds, and electrostatic interactions, control thermodynamic parameters. In the present study, the negative values of ΔG° at all temperatures indicate that HEWL–CTZ is occurring spontaneously in nature (Table 1). Additionally, the positive values of ΔH° and ΔS° indicate that the interaction is derived from entropy rather than enthalpy and is dominated by hydrophobic interactions [2].

### 3.7. Microscopy Imaging

According to all the above-mentioned results, it was demonstrated that CTZ had a dual effect against HEWL amyloid fibrillation. The findings were confirmed by TEM images of HEWL incubated for 8 hr. TEM images indicate a clear amyloid fibril aggregate at pH2, 55 °C, and 600 rpm. Compared to the control, CTZ at 10 µM and 100 µM significantly reduced fibril formation (Figure 8). Furthermore, there is no significant difference in the fibril content and level of inhibition between 10 µM and 100 µM CTZ. Thus, our results suggest that CTZ is an antiaggregating factor. Another point to note is that HEWL alone forms short amyloid structures. However, with the addition of CTZ, the fibril structure has changed into long individual fibrils arranged in large bundles, which differs from the aggregates seen in the absence of CTZ.

### 3.8. Molecular Modeling Studies

The combination of spectroscopic and electron microscopy data indicates that CTZ inhibits HEWL amyloid aggregate formation. Furthermore, HEWL and CTZ interactions at the atomic level were explored through molecular docking to gain insight into structural details. Using Autodock 4.2, information about the putative binding site and mode of interaction was obtained [39,40,41,42,43]. A grid box was created for the site into which CTZ was docked. Next, molecular dynamic simulations were conducted for 5ns for the HEWL–CTZ complex. Molecular docking results indicate that CTZ binds to HEWL with binding energies of −6.58 kcal mol^−1^. HEWL active site is composed of TRP^62^, GLU^35^, ASP^52^, and ALA^107^ [44]. Several previous studies have found that the amino acids 25–33, 55–62, and 107–112 in HEWL tend to form amyloid aggregation [45]. It is important to note that CTZ binds to HEWL regions that contain aggregation-prone regions (APR) and the active site. Thus, binding CTZ to this sensitive APR may stabilize it and prevent it from undergoing conformational changes that may lead to fibrillation [30]. The negative binding energy indicates spontaneous binding between CTZ and HEWL [46]. Molecular docking results are summarized in Table 2. The interaction profiles between CTZ and HEWL is shown in Figure 9. Spectroscopic analysis of HEWL revealed that TRP residues interact with CTZ clearly in its binding pocket. Fluorescence studies indicate that hydrophobic interactions are the primary mode of the CTZ–HEWL interaction, consistent with the positive change in enthalpy obtained. Docking results confirmed that CTZ interacts with HEWL primarily via hydrogen bonds and hydrophobic interactions. A thermodynamic analysis of our fluorescence data indicates that the CTZ displaces structured water molecules at the interface as it binds to HEWL, increasing entropy. Furthermore, CTZ binding reduces the accessibility of solvent to TRP^63^, resulting in a reduction of the fluorescence quenching assay. Figure 9 shows that CTZ has a proximity to the TRP residues (TRP^62^ and TRP^63^) and TYR residues (TYR^53^) of HEWL, explaining why CTZ can quench HEWL’s endogenous fluorescence [47].

### 3.9. Structural Stability of HEWL–CTZ Complex

To study the CTZ–HEWL interaction profile, computational biology methods were used at 300 K [30,39,40]. This computational approach has been used to explain how CTZ could behave against HEWL in vitro. We believe that any ligand will first bind to macromolecules and then alter their conformation, leading to their unfolding or aggregation. MD simulation was carried out using the free HEWL and CTZ–HEWL complex structures obtained from docking studies. Protein aggregation and in vitro fibrillation can be affected by buffer and pH changes. However, in this study, our goal was not to mimic amyloid fibrillation but to study the interaction profile of HEWL and CTZ. Hence, this simulation was not conducted under acidic pH conditions. Furthermore, all factors and charges that could affect protein aggregation were accounted for before docking and simulation.

In a cubic water box, 5 ns MD simulations were performed to evaluate the binding stability of CTZ to HEWL. Through MD simulation, conformational changes in free HEWL and its CTZ-bound form were also examined [48,49,50]. The simulation was performed with Charmm-GUI-NAMD force field [51]. The conformational stability of HEWL upon CTZ binding was investigated using radius of gyration (Rg), root-mean-square deviation (RMSD), root-mean-square fluctuation (Cα RMSF) of the trajectories generated during simulation.

The RMSD is a measure of the equilibration process in a simulation trajectory, as well as the dynamic stability of protein structure after ligand binding. The RMSD values of the backbone atoms of the free and complexed HEWL increase in the first 20 ps because of a protein structure optimization and then the values are essentially stable. Free HEWL backbone atoms achieved equilibrium after 40 ps at 1.21 ± 0.16 Å and did not change significantly during the next 5 ns. A CTZ–HEWL complex reached equilibrium after 40 ps at 1.33 ± 0.20 Å with little change after 100 ps. HEWL has a decrease in stability after binding with CTZ, as the CTZ–HEWL complex achieved equilibrium at a higher RMSD value than free HEWL. As shown in Figure 10A, the RMSD values of CTZ–HEWL are greater than those of HEWL alone, suggesting that CTZ influences HEWL’s structural conformation. Nonetheless, many reports believe this small difference between RMSD values means that ligands have no significant impact on protein structure stability [50]. CTZ binds to HEWL after 100 ps and dissociates at 180 ps, as illustrated in Figure 10A. After 180 ps, the protein returns to fluctuation with just a slight difference from free HEWL at 250 ps. This indicates that CTZ does not significantly destabilize the structure of HEWL. However, to understand this MD simulation in greater detail, longer simulation times are needed.

To evaluate structural compression changes, gyration radius (Rg) diagrams were monitored during the simulation time. Based on the analysis of Rg, HEWL and the CTZ–HEWL complex are compressed throughout the simulation [52]. As shown in Figure 10B, the Rg is considered as an indicator of structural formation in MD simulation [53]. A sudden drop or increase in the Rg profile indicates a structural transformation of the protein. However, it was not observed in the CTZ–HEWL complex [30,54].

Cα RMSF analysis was carried out on each residue in HEWL to determine if CTZ binding influences its flexibility. Figure 10C shows the RMSF of separate residues of HEWL and the CTZ–HEWL complex for the structure in the 5 ns MD run. The RMSF profile of free HEWL is used for comparison with that of CTZ-bound HEWL. The binding of CTZ leads to a significant change in Cα RMSF of residues 3–17, 60–80, and 102–129 (Figure 10 C). It appears that HEWL undergoes minor structural changes when CTZ is bound to it [30]. Cα RMSFs for residues in helix A (LYS^1^, GLU^7^, ALA^10^, ALA^11^, ARG^14^, and HIS^15^), triple-stranded *β*-sheet (ASP^52^ and GLN^57^), large loop (TRP^62^ and TRP^63^), helix C (SER^86^ and ILE^88^), ALA^107^ and VAL^109^ are increased in the CTZ–HEWL complex, highlighting higher flexibility of these regions over HEWL. The highest RMSF values are observed around residues 3–17, 67–76, 102, 118, and 129 indicating that these sections are more flexible. Residues 3–12, 25–43, and 51–65 form the active pocket, which is much less flexible and more stable than the other residues. As compared to free HEWL, residues 3–12 fluctuate more in the HEWL–CTZ complex [47]. As should be noted, the last four residues in HEWL (THR^118^-LEU^129^) fluctuate higher in the HEWL–CTZ complex than in HEWL. Possibly, CTZ binding causes these C-terminal residues to become more stable, hence protecting the folded structure of HEWL [50].

Molecular docking and dynamic simulations were conducted to study the interaction profile between CTZ and HEWL. By determining the amino acid residues that interact with CTZ, we determined whether these amino acids are crucial for protein stability or located within the region of the protein that is prone to aggregation. As a result of CTZ binding, a slight change in the conformation of HEWL took place. The small changes in the confirmation of HEWL could have any fate during the unfolding pathways including hydrophobicity, surface charge, and intermediates. Therefore, full coherence of MD simulation data with in vitro findings is not justified.

Hydrophobic cavities in HEWL accommodate drug molecules, and they are essential to their absorption, metabolism, and transportation. Usually, aromatic drugs are embedded in the bonded zone and the HEWL amino acids TRP^62^, TRP^63^, and Tyr^53^ indicate the stability of the protein–drug system. However, CTZ can only bind to TRP^62^ and TRP^63^, which may be why binding of CTZ inhibits protein aggregation and stabilizes protein structure.

## 4. Discussion

In recent years, the topic of protein aggregation has received a great deal of attention due to its association with systemic diseases and neurological conditions. A million people worldwide suffer from neurodegenerative diseases, such as Alzheimer’s disease (AD), Parkinson’s disease (PD), and Huntington’s disease (HD). In these diseases, large protein aggregates form in the central nervous system. Despite their different sizes and protein structures (primary, secondary, and tertiary), amyloid aggregates associated with different amyloid diseases have similar morphologies [11]. While reports suggest that small organic compounds inhibit fibril formation by a variety of proteins, finding an exact cure remains a challenge. Amyloid aggregates can be formed by macromolecules when they are subjected to pathological conditions. There are several disorders associated with this aggregation process. The amyloid disorder is a serious pathological condition marked by massive protein aggregation. HEWL aggregation has been demonstrated in vitro under specific conditions [2,13]. Since HEWL and HL have very similar structures and folding mechanisms, HEWL can be used to study HL amyloid aggregation. The formation of amyloid fibrils in HEWL was promoted by a high temperature and acidic pH. The presence of amyloid fibrils was then determined using several specific amyloid detection methods. When amyloid aggregates were present in HEWL samples, Thioflavin-T measurements showed high fluorescence intensities, CD spectra showed a negative band at 218 nm, and TEM images showed fibrils formed. In recent years, research has focused on the antiaggregation effect of antihypertensive medications. For this study, CTZ was added to the incubation medium to examine its effects on HEWL amyloid production. The formation of HEWL amyloid is nucleation-dependent, with three distinct phases: nucleation, elongation, and equilibration. Our study examines the effects of CTZ on the equilibrium/saturation phase where mature fibrils form. We investigated the antiamyloidogenic activity of CTZ, a hydrophobic antihypertensive drug.

Our results revealed that CTZ inhibited fibril aggregation at 10 μM and 100 μM concentrations but no significant difference was observed between the concentrations. Additionally, as CTZ concentrations increased, RLS, turbidity, and ANS increased. In addition, since the Thioflavin-T and CD results have not changed significantly, soluble aggregates are likely the primary cause of this increase. A cosolute-dependent change in the solubility model has been reported in recent years. In this model, hydrophobic compounds protect proteins from interactions with aqueous solutions and promote protein prescription, which may lead to increased aggregation [27,55]. Soluble aggregates are expected to be the main components of this prescription because of the lack of change in Thioflavin-T fluorescence. Although we believe that CTZ would not be able to shield the HEWL–solution interactions when present at 10 µM, which is 10-fold lower than the HEWL concentration, this could be true when present at a 1:1 molar ratio with HEWL.

In the present study, 10 μM CTZ significantly decreased the intensity of Thioflavin-T and the number of amyloid fibrils by 76%, while 100 μM of CTZ reduces the Thioflavin-T fluorescence by 74%. This 2% difference between both concentrations is not significant and may be due to the presence of soluble aggregates as suggested by So et al. [27]. The increase in ANS with increasing CTZ does not result from amyloid aggregation, but rather from the formation of soluble aggregates. This is because Thioflavin-T and CD do not exhibit any significant changes in amyloid fibril content. A few previous studies also validate ANS’s binding to soluble aggregate [56,57]. The hydrophobic CTZ was prepared in DMSO and all reaction mixtures contained 1% of DMSO and this was not observed to have a significant effect on protein accumulation [58,59]. To gain further insight into structural details and the mode of interaction between HEWL and CTZ at the atomic level, molecular docking and simulation were conducted. Table 2 lists the amino acids involved in CTZ binding to HEWL. Docking and simulation results indicate that CTZ binds to APR and perhaps exerts antiaggregating effects by stabilizing these regions without destabilizing the HEWL structure.

The limitations of our study are producing aggregates of HEWL at extreme pH and temperature and thus the mechanism and intermediates obtained between folding and aggregation could be different from physiologically aggregated proteins. Further revalidation in cellular or in vivo models would add extra value to our findings. Moreover, several earlier studies reported the formation of amyloid fibrils under acidic conditions combined with agitation [60,61,62]. HEWL denatures under these conditions and forms partially unfolded proteins. These partially unfolded protein give rise to protofibrils that form mature fibrils. According to a previous study, in the presence of acidic conditions and high temperatures, HEWL aggregation follows these steps: Amyloid monomers aggregate into small oligomers of similar size with no nucleation barrier. Nucleation of protofibrils begins when oligomers reach a critical concentration. After protofibrils grow into polymers of oligomers, oligomers are added to their ends. As protofibrils reach a contour length of several hundred nanometers, they self-assemble into longer, stronger, and mature fibrils [63]. We could not explore aggregation properties using MD simulation due to our limited resources, which requires more sophisticated techniques of computational biology.

## 5. Conclusions

In the current study, we examined the antiaggregation effect of CTZ on HEWL amyloid aggregation. CTZ was able to significantly inhibit HEWL amyloid fibrils at 10 μM and 100 μM, but the level of inhibition did not significantly differ between these concentrations. As confirmed by ThT fluorescence assays, TEM micrographs, and CD spectra measurements, CTZ suppresses amyloid fibrillation. As CTZ concentrations were increased, turbidity, RLS, and ANS fluorescence assays increased. This increase may be attributed to the formation of soluble aggregates. Micrographs obtained by TEM showed a reduction in fibril content in the presence of CTZ. In addition, the HEWL amyloid fibrillar exhibits morphological changes in the presence of CTZ. Perhaps the antiaggregation effects of CTZ are a result of its stable binding to HEWL’s APR and blocking the formation of β-sheets. According to fluorescence quenching and molecular docking studies, CTZ binds to HEWL via hydrogen bonds and hydrophobic interactions. Figure 11 illustrates the mechanistic description of HEWL fibrillation and inhibition by CTZ. Our research demonstrates that CTZ can prevent amyloid aggregation in vitro, but in vivo research is essential to validate CTZ’s efficacy and clinical viability. In light of the discovery that diuretics are potent aggregation inhibitors, there is a potential for further expansion of amyloidosis therapeutics.

## Figures and Tables

**Figure 2 ijms-24-03112-f002:**
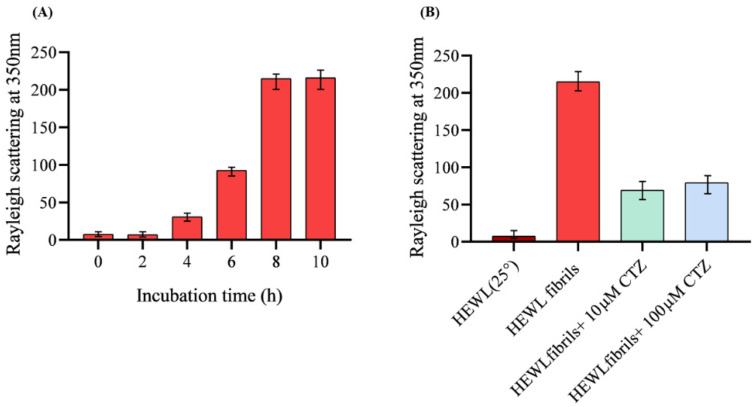
Rayleigh scattering measurements of 10 µM HEWL. (**A**) Represents HEWL fibril formation incubated at pH2.0, 55 °C for 10 h. (**B**) Shows the effect of 10 µM and 100 µM CTZ on HEWL aggregation after 8 h incubation at pH2.0, 55 °C.

**Figure 3 ijms-24-03112-f003:**
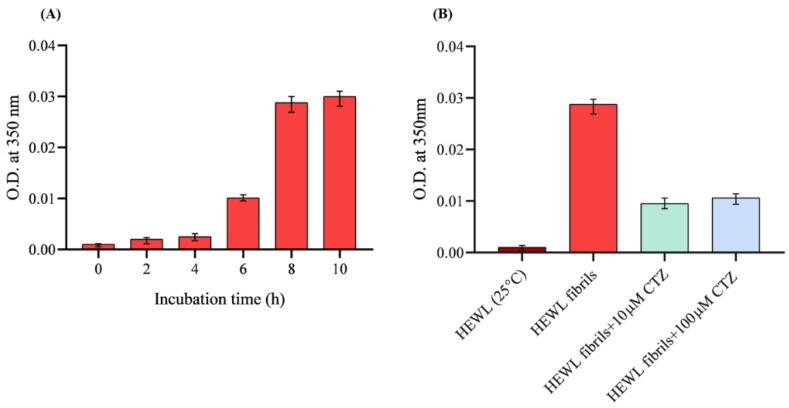
Turbidity measurements by recording the optical density (O.D.) at 350 nm. (**A**) Represents HEWL fibril formation incubated at pH2, 55 °C for 10 h. (**B**) 10 µM HEWL fibrils incubated for 8 h at 55 °C with 600 rpm agitation the presence and absence at 10 µM and 100 µM CTZ.

**Figure 4 ijms-24-03112-f004:**
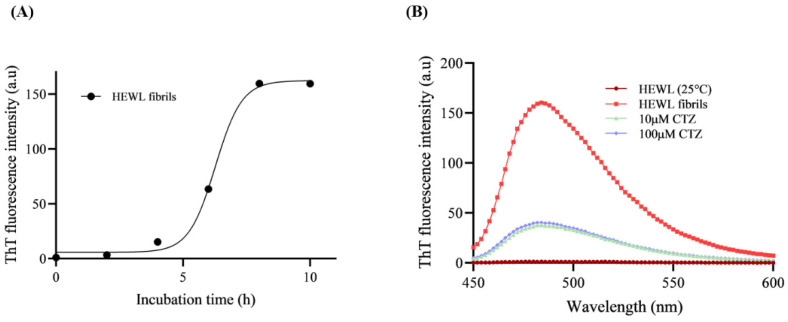
The effect of CTZ toward HEWL fibrillation measured by ThT fluorescence assay. (**A**) ThT fluorescence intensities of 10 µM HEWL monitored at different time points during incubation period. (**B**) ThT fluorescence intansity of 10 µM HEWL in the presence and absence of 10 µM and 100 µM CTZ after 8 h incubation.

**Figure 5 ijms-24-03112-f005:**
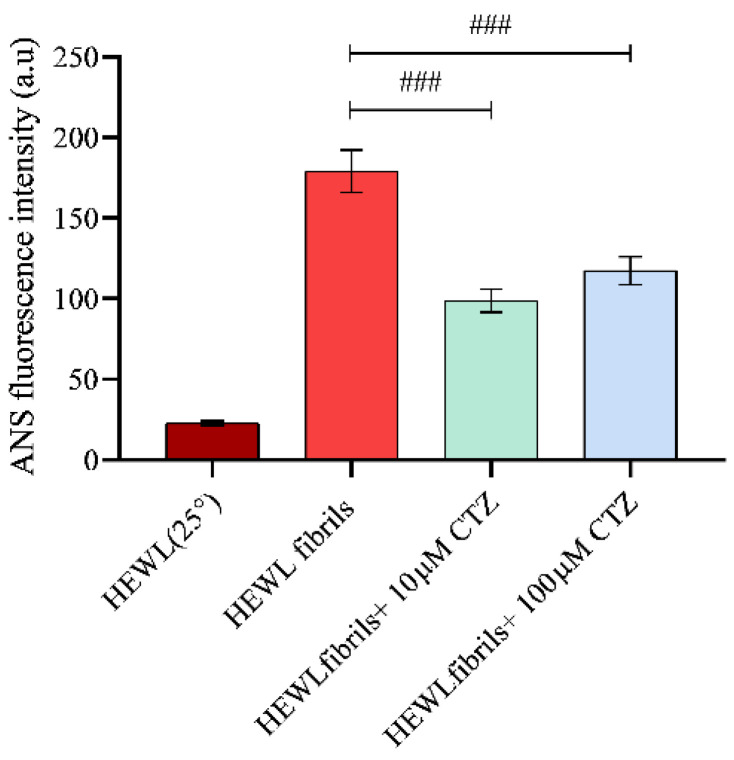
Bar graph representing ANS fluorescence intensity at 480 nm of fibrillated 10 µM HEWL incubated at pH2, 55 °C for 8 h in the presence of 10 µM and 100 µM of CTZ. ^###^
*p* < 0.001, a significant difference from fibrillated HEWL.

**Figure 6 ijms-24-03112-f006:**
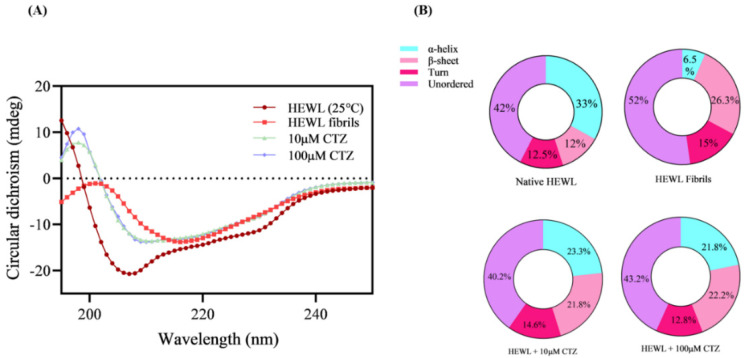
Change in secondary structure in presence of CTZ. (**A**) Far-UV CD spectra of native 10 µM HEWL at 25 °C and HEWL incubated for 10 h at 55 °C with/without 10 µM and 100 µM CTZ. (**B**) Secondary structural contents of HEWL incubated with/without 10 µM and 100 µM CTZ as calculated from BeStSel.

**Figure 7 ijms-24-03112-f007:**
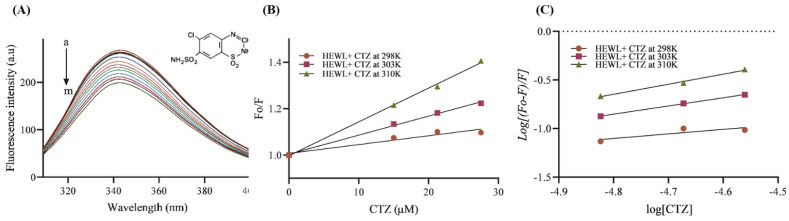
(**A**) Quenching spectra of the HEWL intrinsic fluorescence by increasing amounts of CTZ at 298K. The arrow indicates the increase of the CTZ concentration. C_CTZ_ (a−m) = 0 − 27.5 μM and C_HEWL_ = 5 μM. (**B**) Stern − Volmer plots of HEWL fluorescence quenching by CTZ at 298, 303, and 310K. (C) Plot of modified Stern−Volmer equation regarding the interaction between CTZ and HEWL at 298, 303, and 310K.

**Figure 8 ijms-24-03112-f008:**
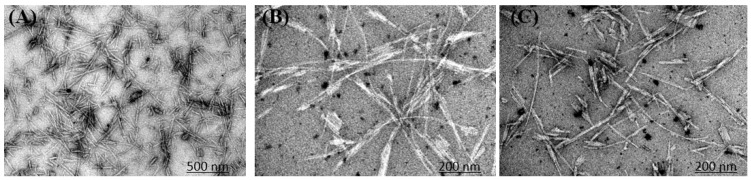
TEM analysis of HEWL fibrils in the absence and presence of CTZ. (**A**) HEWL fibrils. (**B**) HEWL fibrils in presence of 10 µM of CTZ. (**C**) HEWL fibrils in presence of 100 µM of CTZ.

**Figure 9 ijms-24-03112-f009:**
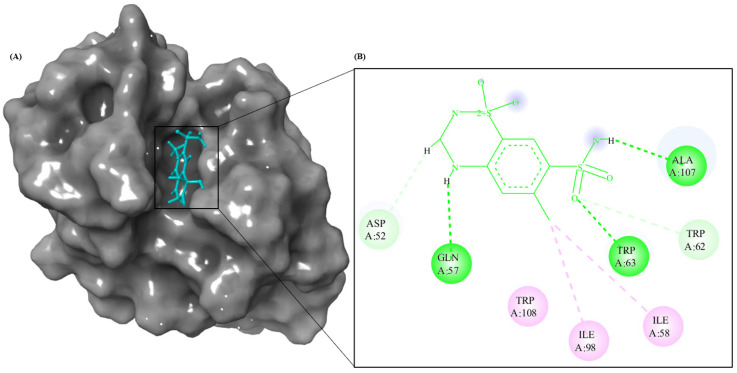
HEWL−CTZ interaction analysis using docking method. (**A**) Panoramic view of the HEWL–CTZ binding mode. (**B**) Interaction profile of HEWL and CTZ. The binding residues in each site were visualized using discovery studio. In the 2D diagram, green color represents hydrogen bonds, pink color represents hydrophobic interactions, and orange color represent electrostatic interactions.

**Figure 10 ijms-24-03112-f010:**
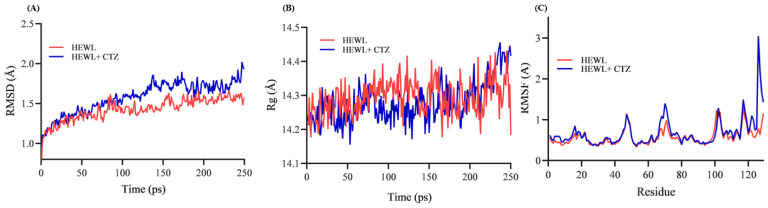
MD simulation equilibration and analysis of the stability of free and bounded HEWL. (**A**) Time evolutions of backbone RMSD of free and CTZ-bounded HEW. (**B**) The Rg of free and CTZ bounded HEW. (**C**) The RMSF of free and CTZ-bound HEW.

**Figure 11 ijms-24-03112-f011:**
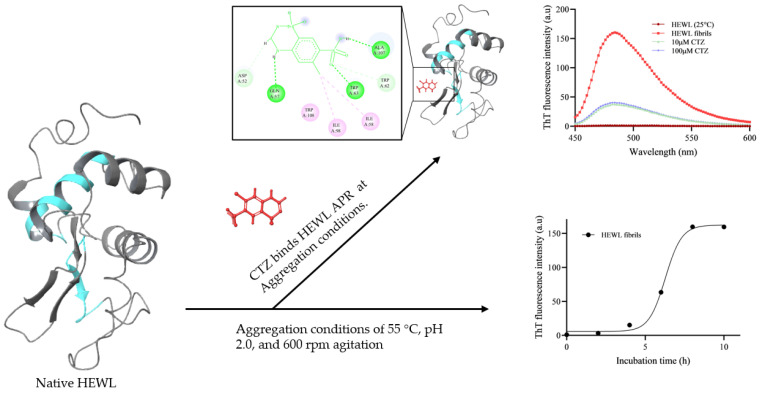
Mechanistic description of HEWL fibrillation and inhibition by CTZ.

**Table 1 ijms-24-03112-t001:** Fluorescence quenching and binding parameters of CTZ–HEWL interactions at different temperatures.

T(K)	K_sv_ (L mol^−1^)	K_q_ (L mol^−1^ s^−1^)	K_b_ (M^−1^)	ΔH° (Kcal mol^−1^)	ΔS° (Kcal mol^−1^ K^−1^)	ΔG° (Kcal mol^−1^)
298.15	3.80 × 10^9^ ± 0.05	3.80 × 10^8^ ± 0.05	1.51 × 10^8^ ± 0.07	36.41	0.12	−0.27
303.15	8.20 × 10^9^ ± 0.0	8.20 × 10^8^ ± 0.09	4.62 × 10^8^ ± 0.11			−0.88
310.15	14.56 × 10^9^ ± 0.17	14.56 × 10^8^ ± 0.17	16.46 × 10^8^ ± 0.13			−1.74

**Table 2 ijms-24-03112-t002:** Amino acid residues of HEWL that interact with CTZ.

Amino Acid Residues	Interactions Involved	Binding Energy (kcal mol^−1^)
ASP^52^	Hydrogen Bonding	−6.58
GLN^57^		
TRP^62^		
TRP^63^		
ALA^107^		
ILE^58^	Hydrophobic Interactions	
ILE^98^		
TRP^108^		

## Data Availability

Not applicable.

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
