# Peer review of "Effect of Antihypertensive Drug (Chlorothiazide) on Fibrillation of Lysozyme: A Combined Spectroscopy, Microscopy, and Computational Study"

_ijms, 2023, doi:10.3390/ijms24043112_

Round 1
Reviewer 1 Report
In the present paper the authors combine different experimental techniques and theoretical approaches to study the influence of Chlorotiazide on fibrillation of HEWL. While the study and its results are interesting for a large scientific communities dealing with amyloid fibrillation the paper needs major revision before publishing. More detailed comments are given below.
1. In the Introduction the authors discuss the role of human lysozme in occurances of different diseases. The study is done on the HEW lysozyme. The authors should discuss the similarities, and differences of both, elaborating on how can their conclusions be significant for drug development on the human lysozme.
2. The experimental part of the study is done at 55oC, and pH=2.0 These conditions are far from physiological ones. The authors should comment on that.
3. Further, the authors compare their experimental findings with computer simulations that was done at 300 K which is very different from experimental one. The justification for such comparison is needed.
4. One of the methods used by the authors is the Turbidity analysis. The authors claim that the increase in the turbidity determined between 350-450 nm is interpreted as the occurrence of fibrils in the solutions. However, this is not true; several studies used this same turbidity measurement for detecting liquid-liquid phase separation of protein solutions where proteins remain in their native form. Additional explanation is needed.
5. When the method of molecular dynamics simulation is described, the authors refer to BS-1 and BS-2, terms that are only explained later in the text. This should be revised. Also, as describing results the authors claim that binding only occurs at these two sites. Where is this claim coming from?
6. The authors rely several of their conclusions on the results of AutoDock. However, it's been known for a while that this program has very poor predicting abilities when it comes to affinities of ligands to protein (see for example Ramirez et al., Int. J. Mol. Sci. 2016, Wang et al., Phys. Chem. Chem. Phys. 2016). Due to the questionable results, the authors should either leave the out completely, or move them to Supplementary section.
7. As already mentioned above, the results of computer simulations are obtained at different temperature than experimental one, however, there are other inconsistencies. Were the buffer molecules taken into account in the MD simulation? The buffer was found to play a major role in the fibrillization process (see for example, Brudar et al., Biomolecules 2019). Also, how was the pH adjusted, were the simulation actually done at pH=2.0?
8. Further, the results of computer simulations seem to contradict the experimental results. While in experiments it was found that the drug slows down the fibrillization process, the results of MD simulations suggest that the changes in the native structure of the HEWL occur in its presence. Since the unfolding of the protein is a precursor for fibrils to be formed, this contradicts with experimental findings. The autrhos shoule elaborate on that.
9. The authors use several techniques to detect fibrils, so there is no question whether or not they form in the solution. However, similarly to turbidity analysis, ANS fluorescence is also not reliable indicator for detecting hydrophobic patches. It is well known that ANS can also bind to the external sites of the protein, forming ion-pair formations (see for example Gasymov et al., Biochim. Biophys. Acta 2008). The authors should therefore be more careful making the conclusions based on its fluorescence.
10. Minor comment: several typos exist in the text that should be corrected.
Author Response
Response to Reviewer-1
Comment 1# In the Introduction the authors discuss the role of human lysozme in occurances of different diseases. The study is done on the HEW lysozyme. The authors should discuss the similarities, and differences of both, elaborating on how can their conclusions be significant for drug development on the human lysozyme.
Response: We appreciate the reviewer's comments. We have added these points and details in our revised our manuscript.
Comment 2# The experimental part of the study is done at 55oC, and pH=2.0 These conditions are far from physiological ones. The authors should comment on that.
Response: We appreciate the reviewer's comments and valid concern. These non-physiological conditions is used to induce amyloid aggregation in vitro and mimic the abnormal in vivo conditions that lead to abnormal aggregates (Wu, J. W., Liu, K. N., How, S. C., Chen, W. A., Lai, C. M., Liu, H. S., Hu, C. J., & Wang, S. S. S. (2013). Carnosine’s effect on amyloid fibril formation and induced cytotoxicity of lysozyme. PLoS ONE, 8(12). Moreover, proteins found in amyloidosis are not of any specific conformation and restructured during the process of unfolding and aggregation could attain any possible conformations possibly at higher temperatures and drastic low pH. Therefore, characterizations of conformations obtained in these conditions would help to design inhibitors against protein misfolding diseases.
Comment 3# Further, the authors compare their experimental findings with computer simulations that was done at 300 K which is very different from experimental one. The justification for such comparison is needed.
Response: We thank the reviewer for his/her comment. Our study of chlorothiazide-HEWL interaction profile was conducted using computational biology methods (docking & simulation) to predict the possible interaction of CTZ with lysozyme and getting a clue for in vitro experimentations. We also believes that any ligand will bind to the macromolecules first and thereafter change in the conformations leading to unfolding or aggregation take place. Therefore, binding and aggregation measurement could co-relate each other. However, thermodynamics parameters derived from binding studies could not be extrapolate and defined at higher temperatures.
Comment 4# One of the methods used by the authors is the Turbidity analysis. The authors claim that the increase in the turbidity determined between 350-450 nm is interpreted as the occurrence of fibrils in the solutions. However, this is not true; several studies used this same turbidity measurement for detecting liquid-liquid phase separation of protein solutions where proteins remain in their native form. Additional explanation is needed.
Response: As our studies focused on protein binding and aggregation, we usually measure turbidity of protein solution between 350-650 nm absorbance. This is the basic techniques to detect large aggregates and routinely measured in our lab. Moreover, this technique is used to measure aggregates and not specific fibrillation which can be verified only after ThT, congo red and other dyes specific measurement. We have modified the interpretation in result section. Turbidity of protein aggregates at 350 nm has been previously measured in several studies (Chaturvedi, S. K. et al. Unraveling comparative anti-amyloidogenic behavior of pyrazinamide and D-Cycloserine: A mechanistic biophysical insight. PLoS One 10, (2015); Bag, S., Chaudhury, S., Pramanik, D., DasGupta, S. & Dasgupta, S. Hydrophobic tail length plays a pivotal role in amyloid beta (25–35) fibril–surfactant interactions. Proteins: Structure, Function and Bioinformatics 84, (2016); Khan, J. M. et al. SDS can be utilized as an amyloid inducer: A case study on diverse proteins. PLoS One 7, (2012).
Comment 5# When the method of molecular dynamics simulation is described, the authors refer to BS-1 and BS-2, terms that are only explained later in the text. This should be revised. Also, as describing results the authors claim that binding only occurs at these two sites. Where is this claim coming from?
Response: We are sorry for the overinterpretation of our MD simulation study. MD simulation was carried out to evaluate the binding stability of CTZ to HEWL. Through MD simulation, conformational changes in free HEWL and its CTZ-bound form were examined.
Comment 6# The authors rely several of their conclusions on the results of AutoDock. However, it's been known for a while that this program has very poor predicting abilities when it comes to affinities of ligands to protein (see for example Ramirez et al., Int. J. Mol. Sci. 2016, Wang et al., Phys. Chem. Chem. Phys. 2016). Due to the questionable results, the authors should either leave the out completely, or move them to Supplementary section.
Response: We thank the reviewer for his/her comment. It is important to note that in our study, we used AutoDock tools as an initial step for molecular dynamic simulation. We aimed to dock chlorothiazide in HEWL and identify binding residues. Several recent publications have also used this tool [1-5]. Our results and conclusion are not solely based on docking and simulation studies but also validated by specific spectroscopic and microscopic techniques. Due to concern of learned referee, we have restructured our conclusion that shows comparison of our in vitro data with silico modeling.
Comment 7 # As already mentioned above, the results of computer simulations are obtained at different temperature than experimental one, however, there are other inconsistencies. Were the buffer molecules taken into account in the MD simulation? The buffer was found to play a major role in the fibrillization process (see for example, Brudar et al., Biomolecules 2019). Also, how was the pH adjusted, were the simulation actually done at pH=2.0?
Response: We thanks the reviewer for his/her comments. The buffer and pH change, as mentioned in the reference attached, can affect protein aggregation; however, our goal was not to mimic amyloid fibrillation, but to study the interaction profile of HEWL-Chlorothiazide, and all factors and charges that could affect protein aggregation were accounted before docking and simulation.
Comment 8# Further, the results of computer simulations seem to contradict the experimental results. While in experiments it was found that the drug slows down the fibrillization process, the results of MD simulations suggest that the changes in the native structure of the HEWL occur in its presence. Since the unfolding of the protein is a precursor for fibrils to be formed, this contradicts with experimental findings. The authors should elaborate on that.
Response: Thanks to the reviewer for his/her comments. We want to clarify that molecular docking and dynamic simulation were conducted to study the interaction profile of chlorothiazide and HEWL. By determining the amino acid residues that interact with chlorothiazide, we determine if these amino acids are crucial for protein stability or located within the region of the
protein that is prone to aggregation. As a result of chlorothiazide binding to some regions of proteins a slight change in the conformation of HEWL took place. The small changes in confirmation of HEWL could have any fate during the unfolding pathways including hydrophobicity, surface charge, and intermediates.
Comment 9# The authors use several techniques to detect fibrils, so there is no question whether or not they form in the solution. However, similarly to turbidity analysis, ANS fluorescence is also not reliable indicator for detecting hydrophobic patches. It is well known that ANS can also bind to the external sites of the protein, forming ion-pair formations (see for example Gasymov et al., Biochim. Biophys. Acta 2008). The authors should therefore be more careful making the conclusions based on its fluorescence.
Response: We appreciate the reviewer's comment but would like to point out that ANS fluorescence measurements have been extensively used to study protein aggregation. It was used in our study to evaluate the conformational changes in HEWL during temperature and pH induced unfolding. This is one of the critical hallmarks of amyloid aggregation that is not present in natively folded proteins (hen, X. et al. Inhibition of Lysozyme Amyloid Fibrillation by Silybin Diastereoisomers: The Effects of Stereochemistry. ACS Omega 6, (2021). Amyloid specific dye ThT and cross-β-sheet structure as observed in CD spectroscopy and TEM analysis reaffirm the fibrillation process and their inhibition.
Comment 10# Minor comment: several typos exist in the text that should be corrected.
Response: We thank the reviewer for his/her comments. It has been fixed throughout the manuscript.
References
- Patel, K. Parmar, and M. Das, “Inhibition of insulin amyloid fibrillation by Morin hydrate,” Int J Biol Macromol, vol. 108, 2018, doi: 10.1016/j.ijbiomac.2017.11.168.
- S. S. Narang, S. Shuaib, and B. Goyal, “Molecular insights into the inhibitory mechanism of rifamycin SV against β2–microglobulin aggregation: A molecular dynamics simulation study,” Int J Biol Macromol, vol. 102, 2017, doi: 10.1016/j.ijbiomac.2017.04.086;
- F. J. Kelutur and R. Mustarichie, “Molecular docking of the potential compound from cocoa shells (theobroma cacao l.) against androgen receptor as anti- alopecia,” Journal of Global Pharma Technology, vol. 12, no. 9, 2020.;
- M. M. Julie, T. Prabu, F. B. Asif, and S. Muthu, “In silico drug evaluation and drug research of bioactive molecule methyl 4-bromo-2-fluorobenzoate,” Ankara Universitesi Eczacilik Fakultesi Dergisi, vol. 45, no. 2, 2021, doi: 10.33483/jfpau.897720.;
- E. Akaho, “Epigenetic Drugs, and Their Virtual Screening Study Retrieved from ZINC Database Along with an AutoDock Study of the Best Inhibitor,” in Current Aspects in Pharmaceutical Research and Development Vol. 7, 2022. doi: 10.9734/bpi/caprd/v7/3114e].

Reviewer 2 Report
The manuscript presented the effect of chlorothiazide (CTZ) on the fibrillation of the lysozyme. The authors used a combination of spectroscopic, microscopic, and computational approaches. They employed HEWL as a model protein for fibrillation. HEWL aggregation and the formation of amyloid structures under protein misfolding conditions were confirmed by increased turbidity and Rayleigh light scattering, as well as by TEM and Tht fluorescence. The authors observed the dose-dependent inhibition effect of CTZ on HEWL aggregation, confirmed by turbidity, Rayleigh light scattering, Tht fluorescence, and TEM. In addition, ANS fluorescence measurement revealed a decrease in the HEWL protein in the presence of CTZ. The computational results found two binding sites of CTZ, each with different binding energy, suggesting the reason for the anti-amyloidogenic activity. To obtain these conclusions, the authors used a number of different techniques (10) and received a number of results. This is the main added value of the manuscript. However, the main weakness is the insufficient description of the topic and unclear, sometimes chaotic interpretation of the results. This makes otherwise remarkable results unclear. The Introduction is too brief and does not provide sufficient insight into the topic. For example, it is not described why diuretics have been used to reduce amyloid aggregation, what kind of diuretics have been used to prevent amyloid-related disease and what the difference is between their study and previous studies (e.g., 13,16). The Methods are described adequately, allowing the repetition of the experiments. However, the Results and Discussion parts lack a more comprehensive analysis. Especially Discussion part lacks the intuitive connection of all measured and calculated results into one logical unit, which would be compared with other available results. Conclusions are insufficiently and sometimes incorrectly described given the number of interesting results from which they derive (e.g., sentence: “10µM CTZ, can effectively reduce HEWL aggregation, while 100µM CTZ can cause the opposite”). In conclusion, in this state, the manuscript requires major revisions.
My additional comments:
Manuscript: Professional English proofreading is needed.
Abstract: HEWL and CD abbreviations are not defined.
Figure 3: O.D. is not defined.
Figures 2,3,5: The bottom of the error bar is not visible.
Results and Discussion: The dual effect of CTZ is not adequately described and can be a source of data misinterpretation. Why is it not mentioned in the Abstract and the Introduction? Some parts of Results and Discussion are more suitable for the Introduction (e.g., “HEWL is a globular protein composed of two domains, the α-domain, and the β-domain. α-domain is composed of five helixes.”)
References: Why do authors cite preprints of articles that were published a long time ago?
Author Response
Response to Reviewer-2
Comment 1# The Introduction is too brief and does not provide sufficient insight into the topic. For example, it is not described why diuretics have been used to reduce amyloid aggregation, what kind of diuretics have been used to prevent amyloid-related disease.
Response: We thank the reviewer for his comments. It has been fixed in the manuscript and added (highlighted in red) to the introduction part of our revised manuscript.
Comment 2# What the difference is between their study and previous studies (e.g., 13,16).
Response: Previous studies measured the anti-aggregation effects of diuretics (Indapamide & Hydrochlorothiazide) in wet lab experiments. However, in our paper, we aimed to assess the anti-aggregation effects of diuretics as well as their interaction profiles with chlorothiazide and HEWL using spectroscopic, microscopic and computational biology. Earlier reports also did not provide any information on binding and thermodynamics of diuretics with HEWL. Moreover, our aggregation studies were optimized under different unfolding conditions of HEWL which could mimic similar to the conformation attained in amyloidosis. Finally, simulation studies were not included in previous studies and thus added more values to our manuscript
Comment 3# The Methods are described adequately, allowing the repetition of the experiments. However, the Results and Discussion parts lack a more comprehensive analysis. Especially Discussion part lacks the intuitive connection of all measured and calculated results into one logical unit, which would be compared with other available results.
Response: We thank the reviewer for his/her comments. We have tried our best to improve the result and discussion part to make a good connection between obtained results and available scientific literature.
Comment 4# Conclusions are insufficiently and sometimes incorrectly described given the number of interesting results from which they derive (e.g., sentence: “10μM CTZ, can effectively reduce HEWL aggregation, while 100μM CTZ can cause the opposite”). In conclusion, in this state, the manuscript requires major revisions.
Response: We thank the learned reviewer for his/her critical comments. After careful analysis of our results, we would like to apologize for our misinterpretation of results. The above statement has been amended as per the obtained data in the revised manuscript.
Additional comments
Comment 5# Manuscript: Professional English proofreading is needed.
Response: We thank the reviewer for his/her comments. It has been fixed in the manuscript using Grammarly software.
Comment 6# Abstract: HEWL and CD abbreviations are not defined.
Response: We have fixed it in the revised manuscript.
Comment 7# Figure 3: O.D. is not defined. Response: We have fixed it in the revised manuscript.
Comment 8# Figures 2,3,5: The bottom of the error bar is not visible.
Response: Thank you for the reviewer's comments. We have fixed it in the revised manuscript
1- Results and Discussion: The dual effect of CTZ is not adequately described and can be a source of data misinterpretation. Why is it not mentioned in the Abstract and the Introduction? Some parts of Results and Discussion are more suitable for the Introduction (e.g., “HEWL is a globular protein composed of two domains, the α-domain, and the β-domain. α-domain is composed of five helixes.”).
Response: We thank the reviewer for his/her comments. It has been fixed in the revised manuscript. We agree with the reported dual effect of CTZ which was actually data misinterpretation. After careful analysis of our results, we would like to apologize for our misinterpretation of results. The statement has been amended as per the obtained data in the revised manuscript.
2- References: Why do authors cite preprints of articles that were published a long time ago?
Response: We thank the reviewer for his/her comment. We have fixed it in the revised manuscript.

Round 2
Reviewer 1 Report
Even though the authors addressed all the comments they did not give sufficient justification or modify the text of the paper appropriately to clear the points. I therefore recommend major revision, and request the authors to modify the paper accordingly to the comments.

Author Response
Response to Reviewer- 1
The authors have addressed some of my concerns, however, further modifications of the paper are in my opinion necessary for the paper to be suitable for publication. The authors should clearly state the limitations of their study. Major revision is necessary!
Comment 1# The experimental part of the study is done at 55oC, and pH=2.0 These conditions are far from physiological ones. The authors claim that they were only interested in obtaining the unfolded protein. However, the interactions between unfolded proteins that lead to amyloid fibril formation still depend on the conditions in the solution, such as pH, and temperature. The authors should clearly state these limitation in the paper.
Response: We thank the reviewer for his/her comments. We have added the limitation of our study (highlighted in red) to the discussion part of our revised manuscript, page 14.
Comment 2# Further, the authors compare their experimental findings with computer simulations that was done at 300 K which is very different from experimental one. The justification for such comparison is needed.
The authors should add their explanation into the paper and state the consequent limitations of their conclusions.
Response: We thank the reviewer for his/her comments. We have discussed this in the result and discussion section of our revised manuscript (highlighted in blue). We have also added the limitation of our study and resources, page 12We have also added specific information about the simulation box in methods part, page 5.
Comment 3# One of the methods used by the authors is the Turbidity analysis. The authors claim that the increase in the turbidity determined between 350-450 nm is interpreted as the occurrence of fibrils in the solutions. However, this is not true; several studies used this same turbidity measurement for detecting liquid-liquid phase separation of protein solutions where proteins remain in their native form. Additional explanation is needed.
In their answer the authors are just repeating the same argument that is not valid. They should clearly state in the paper that the turbidity measurement is used to identify large aggregates that are possible fibrils in their case, judging from the results of other techniques. The same should be done for ANS (comment 9).
Response: We thank the reviewer for his comments. It has been fixed and (highlighted in purple) amended in the revised manuscript (page 6 and 8).
Comment 4# When the method of molecular dynamics simulation is described, the authors refer to BS-1 and BS-2, terms that are only explained later in the text. This should be revised. Also, as describing results the authors claim that binding only occurs at these two sites. Where is this claim coming from?
The text modification is needed in this case!!!
Response: As mentioned in the earlier revised version too, we are sorry for the overinterpretation of our MD simulation study. The preferential binding of drugs on BS-1 and BS-2 sites could not justify our in-vitro findings, hence this statement has been amended. MD simulation was carried out to evaluate the binding stability of CTZ to HEWL.
Comment 5 # As already mentioned above, the results of computer simulations are obtained at different temperature than experimental one, however, there are other inconsistencies. Were the buffer molecules taken into account in the MD simulation? The
buffer was found to play a major role in the fibrillization process (see for example, Brudar et al., Biomolecules 2019). Also, how was the pH adjusted, were the simulation actually done at pH=2.0?
This should be explained in the paper. Also, how was buffer in particular taken into account?
Response: We thank the reviewer for his/her comments. It has been explained in the manuscript and added (highlighted in orange) to the result section of MD simulation, page 12.
Comment 6# Further, the results of computer simulations seem to contradict the experimental results. While in experiments it was found that the drug slows down the fibrillization process, the results of MD simulations suggest that the changes in the native structure of the HEWL occur in its presence. Since the unfolding of the protein is a precursor for fibrils to be formed, this contradicts with experimental findings. The authors should elaborate on that.
This has to be commented in the paper!!!
Response: Thanks to the reviewer for his/her comments. We want to clarify that molecular docking and dynamic simulation were conducted to study the interaction profile of chlorothiazide and HEWL. By determining the amino acid residues that interact with chlorothiazide, we determine if these amino acids are crucial for protein stability or located within the region of the protein that is prone to aggregation. As a result of chlorothiazide binding to some protein regions, a slight change in the conformation of HEWL took place. The small changes in confirmation of HEWL could have any fate during the unfolding pathways including hydrophobicity, surface charge, and intermediates.
We thank the reviewer for his comments. It has been fixed in the manuscript and added (both highlighted in green) results part of our revised manuscript, page 13.

Reviewer 2 Report
The authors revised the manuscript due to all my comments and answered all questions. Since the manuscript has significantly improved, I endorse the publication in this form.
Author Response
Comments and Suggestions for Authors
The authors revised the manuscript due to all my comments and answered all questions. Since the manuscript has significantly improved, I endorse the publication in this form.
Response: Thank you so much for critical evaluation and approval for publication for our submitted manuscript.
Round 3
Reviewer 1 Report
The authors have addressed all my concerns, and appropriately modified the manuscript. I believe it is now suitable for publication.